# EgoTaskQA: Understanding Human Tasks in Egocentric Videos

**Baoxiong Jia**[1,2†]
baoxiongjia@ucla.edu

**Ting Lei**[2,3†]
ting_lei@pku.edu.cn

**Song-Chun Zhu**[2,3,4]
sczhu@bigai.ai

**Siyuan Huang**[2]
syhuang@bigai.ai

[1]UCLA Center for Vision, Cognition, Learning, and Autonomy (VCLA)
[2] Beijing Institute for General Artificial Intelligence (BIGAI)
[3] Institute for Artificial Intelligence, Peking University
[4] Department of Automation, Tsinghua University
https://sites.google.com/view/egotaskqa

## Abstract

Understanding human tasks through video observations is an essential capability of intelligent agents. The challenges of such capability lie in the difficulty of generating a detailed understanding of situated actions, their effects on object states (*i.e.*, state changes), and their causal dependencies. These challenges are further aggravated by the natural parallelism from multi-tasking and partial observations in multi-agent collaboration. Most prior works leverage action localization or future prediction as an *indirect* metric for evaluating such task understanding from videos. To make a *direct* evaluation, we introduce the EgoTaskQA benchmark that provides a single home for the crucial dimensions of task understanding through question-answering on real-world egocentric videos. We meticulously design questions that target the understanding of (1) action dependencies and effects, (2) intents and goals, and (3) agents' beliefs about others. These questions are divided into four types, including descriptive (what status?), predictive (what will?), explanatory (what caused?), and counterfactual (what if?) to provide diagnostic analyses on *spatial, temporal, and causal* understandings of goal-oriented tasks. We evaluate state-of-the-art video reasoning models on our benchmark and show their significant gaps between humans in understanding complex goal-oriented egocentric videos. We hope this effort will drive the vision community to move onward with goal-oriented video understanding and reasoning.

## 1 Introduction

The study of human motion perception has suggested that humans perceive motion as goal-directed behaviors rather than plain pattern movements [1–3]. Developmental psychologists [4] categorized such an ability into two distinct mechanisms: (1) action-effect associations that the desired effects activate the corresponding action; and (2) simulative procedures, which argues that goal attribution comes from planning under the rational action principle in others' shoes. Both mechanisms require detailed knowledge of **action dependencies and effects**, agent's **intents and goals** and **beliefs about other agents**. With such knowledge playing crucial roles in human cognitive development, learning them from visual observation is pivotal for building more intelligent agents.

---

[†]Work done during internship at BIGAI.

36th Conference on Neural Information Processing Systems (NeurIPS 2022) Track on Datasets and Benchmarks.

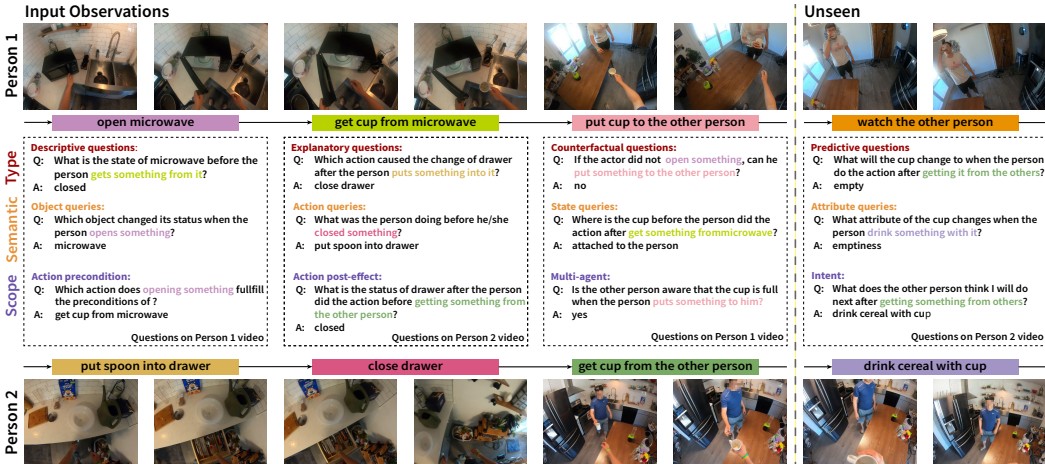

Figure 1: An overview of EgoTaskQA. We show an illustrative scenario where two subjects collaborate to make and drink cereal. Based on egocentric observations, we generate questions on these seen or unseen video intervals with different question types, targeting semantics and question scopes. Note that we use both direct (*e.g.*, when the person open something...) and indirect (*e.g.*, the action before getting something...) references on actions and objects, where the same color indicates the same referred actions (best viewed in color).

Taking a closer look at how humans learn from interacting with the world, we locate objects, change their positions and manipulate them in various ways, all presumably under visual control from an egocentric perspective [5]. This unique first-person experience provides essential visual cues for human attention and goal-oriented task understanding. Moreover, egocentric perception naturally reflects how humans reason and perform in a partially observable environment, making it the most available learning source for learning actions, tasks [6], and belief modeling [7]. The past few years have witnessed significant progress in egocentric video understanding, especially action recognition and future anticipation [8–13]. However, these two tasks merely cover the tip of the iceberg, considering how humans learn from visual observations to obtain knowledge for more profound tasks such as learning world models, planning for desired goals, and building beliefs about others. With their essential roles in human cognitive development, we urge the need for a benchmark that addresses these missing dimensions in egocentric activity understanding.

Hence, we present EgoTaskQA, a challenging egocentric, goal-oriented video question-answering benchmark based on the LEMMA dataset [11]. The LEMMA dataset collects egocentric videos in goal-oriented and multi-agent collaborative activities with fine-grained action and task annotation. By extending the LEMMA dataset with annotations consisting of object status, human-object and multi-agent relationships, and causal dependency structures between actions, we design questions that target three specific scopes: (1) actions with world state transitions and their dependencies, (2) agents' intents and goals in task execution, and (3) agents' belief about others in collaboration to provide an in-depth evaluation metric for task understanding. These questions are procedurally generated within four types: **descriptive**, **predictive**, **explanatory**, and **counterfactual**, to systematically test models' capabilities over **spatial**, **temporal**, and **causal** domains of goal-oriented task understanding. To avoid spurious correlations in questions, we include both direct and indirect references to actions and objects. We further balance the answer distribution by the reasoning type of questions and carefully design benchmarking train/test splits to provide a systematic test on goal-oriented reasoning and indirect reference understanding; see Fig. 1 for an example and more details in Sec. 3.

As shown in Tab. 1, EgoTaskQA complements existing video reasoning benchmarks on various dimensions. With models exhibiting large performance gaps compared with humans, we devise diagnostic experiments to reveal both the easy and challenging spots in our benchmark. We hope such designs and analyses will foster new insights into goal-oriented activity understanding.

**Contributions** In summary, our main contributions are three-fold:

- We extend the LEMMA dataset with annotations of object status, human-object and multi-agent relationships to facilitate egocentric activity understanding. We further generate causal dependency structures between actions to provide ground truth for procedural task understanding.

Table 1: A comparison between EgoTaskQA and existing video question-answering benchmarks. We use "world" for world model-related information, including action preconditions, post-effects, and dependencies. We use FPV as short for egocentric and TPV for third-person-view videos. We use MC as short for multiple-choice question-answering, and OP for open-answer question-answering.

| Dataset | Video | | Question Scope | | | Question type | | | | Answer Type | # questions |
|---|---|---|---|---|---|---|---|---|---|---|---|
| | View | Real-world | World | Intents & Goals | Multi-agent | Descriptive | Predictive | Explanatory | Counterfactual | | |
| MarioQA [42] | TPV | ✗ | ✓ | ✗ | ✗ | ✓ | ✗ | ✓ | ✗ | OP | 188K |
| Pororo-QA [43] | TPV | ✗ | ✗ | ✗ | ✗ | ✓ | ✓ | ✓ | ✗ | MC | 9K |
| CLEVRER [44] | TPV | ✗ | ✗ | ✗ | ✗ | ✓ | ✓ | ✓ | ✓ | OP+MC | 282K |
| Env-QA [45] | FPV | ✗ | ✓ | ✗ | ✗ | ✓ | ✗ | ✗ | ✗ | OP | 85K |
| MovieQA [46] | TPV | ✓ | ✗ | ✗ | ✗ | ✓ | ✗ | ✓ | ✗ | MC | 14K |
| Social-IQ [47] | TPV | ✓ | ✗ | ✓ | ✓ | ✓ | ✗ | ✓ | ✗ | MC | 7.5K |
| TVQA [48] | TPV | ✓ | ✗ | ✗ | ✗ | ✓ | ✗ | ✓ | ✗ | MC | 152.5K |
| TVQA+ [49] | TPV | ✓ | ✗ | ✗ | ✗ | ✓ | ✗ | ✓ | ✗ | MC | 29.4K |
| MSVD-QA [50] | TPV | ✓ | ✗ | ✗ | ✗ | ✓ | ✗ | ✗ | ✗ | OP | 50.5K |
| MSRVTT-QA [50] | TPV | ✓ | ✗ | ✗ | ✗ | ✓ | ✗ | ✗ | ✗ | OP | 243K |
| Video-QA [51] | TPV | ✓ | ✗ | ✗ | ✗ | ✓ | ✗ | ✗ | ✗ | OP | 175K |
| ActivityNet-QA [52] | TPV | ✓ | ✗ | ✗ | ✗ | ✓ | ✗ | ✗ | ✗ | OP | 58K |
| TGIF-QA [53] | TPV | ✓ | ✗ | ✗ | ✗ | ✓ | ✗ | ✗ | ✗ | MC | 165.2K |
| How2QA [54] | TPV | ✓ | ✗ | ✗ | ✗ | ✓ | ✗ | ✗ | ✗ | MC | 44K |
| HowToVQA69M [55] | TPV | ✓ | ✗ | ✗ | ✗ | ✓ | ✗ | ✗ | ✗ | OP | 69M |
| AGQA [56] | TPV | ✓ | ✗ | ✗ | ✗ | ✓ | ✗ | ✗ | ✗ | OP | 3.6M |
| NExT-QA [57] | TPV | ✓ | ✗ | ✓ | ✗ | ✓ | ✓ | ✓ | ✗ | OP+MC | 52K |
| STAR [58] | TPV | ✓ | ✗ | ✗ | ✗ | ✓ | ✓ | ✗ | ✗ | MC | 60K |
| EgoVQA [59] | FPV | ✓ | ✗ | ✗ | ✓ | ✓ | ✗ | ✗ | ✗ | OP+MC | 520 |
| EgoTaskQA (Ours) | FPV | ✓ | ✓ | ✓ | ✓ | ✓ | ✓ | ✓ | ✓ | OP | 40K |

- We construct a balanced video question-answering benchmark, EgoTaskQA, to measure models' capability in understanding action dependencies and effects, intents and goals, as well as beliefs in multi-agent scenarios. We procedurally generate four challenging types of questions (descriptive, predictive, explanatory, and counterfactual) with both direct and indirect references for our benchmark and potential research on video-grounded compositional reasoning.
- We devise challenging benchmarking splits over EgoTaskQA to provide a systematic evaluation of goal-oriented reasoning and indirect reference understanding. We experiment with various state-of-the-art video reasoning models, show their performance gap compared with humans, and Acknowledgementanalyze their strengths and weaknesses to promote future research on goal-oriented task understanding.

## 2   Related Work

**Action as Inverse Planning**   Action understanding has been seen as an inverse planning problem on agents' mental states [14, 15]. Early studies formulate it as reasoning on the first-order logic formulae that describes actions' preconditions and post-effects [16, 17]. This symbolic formalism is later paired with domain-specific language and algorithms to become mainstays in robotics planning [18, 19]. In computer vision, similar attempts have been made to link visual observations with world states and actions [20–22]. Various methods treated actions as transformations on images to solve action-state recognition [23–27] and video prediction [28–30]. With the emerging interest in language-grounded understanding, Zellers et al. [31] proposed PIGLeT to study the binding between images, world states, and action descriptions. Padmakumar et al. [32] further studies the problem of language understanding and task execution by designing an intelligent embodied agent that can chat during task execution. However, these works are mostly limited to atomic actions, missing the important action dependency in task execution. To tackle this problem, instructional videos [33–36] are studied with its goal-oriented multi-step activities. In these videos, external knowledge [37, 38] can be used as guidance for advanced tasks like temporal dynamics learning [39] and visually grounded planning [40, 41]. Unfortunately, these videos highlight the instructions and include no task-level noise, which is much simpler than the partially observable, highly paralleled, multi-agent environment that humans learn from and as presented in our benchmark. These complexities make the goal-oriented action understanding a challenging task remaining to be solved.

**Egocentric Vision**   Egocentric vision offers a unique perspective for actively engaging with the world. Aside from traditional video understanding tasks like video summarization [60, 61], activity recognition [62–64] and future anticipation [65–69], egocentric videos provide fine-grained information for tasks like human-object interaction understanding [70–76] and gaze/attention prediction [77, 10]. With its natural reflectance of partial observability, egocentric videos are also used for social understanding tasks such as joint attention modeling [78, 79], perspective taking [80, 81] and communicative modeling [82, 7]. However, with various egocentric datasets curated over the

last decade [8, 60, 9], data and detailed annotations for human tasks are still largely missing. Large-scale daily lifelog datasets like EPIC-KITCHENS [12] and Ego4D [13] cover certain aspects of action-dependencies, effects, and social scenarios in their recordings, but are unsuitable for detailed annotation due to their size. The other stream of datasets collects activities by providing coarse task instructions to both single actor [83] and multiple agent collaborations [11]. They annotate tasks and compositional actions to reveal agents' execution and collaboration process for multi-step goal-directed tasks. Despite all the preferred characteristics of these goal-oriented activity videos, none of them successfully addressed action-dependencies and effects, nor multi-agent belief modeling.

**Video Question-Answering Benchmarks**   Visual question-answering can be designed to evaluate a wide spectrum of model capabilities, spanning from visual concept recognition and spatial relationship reasoning [84–87], abstract reasoning [88–93], to common sense reasoning [94, 95]. In the temporal domain, synthetic environments are used for questions that involve simple action-effect reasoning [42, 43]. Crowdsourced videos [53, 52, 48, 55] are used for collecting questions on basic spatial-temporal reasoning capabilities like event counting [53], grounding [49], and episodic memory [13]. Recent advances in video question-answering aim for more profound reasoning capabilities. Gao et al. [45] leverages an indoor synthetic environment to generate questions on spatial relationships and simple action-effect reasoning from an egocentric perspective. Xiao et al. [57] designs NExT-QA containing questions about knowledge of the past, present, and future on both temporal and causal domains. Grunde-McLaughlin et al. [56] programmatically generates questions for compositional spatial-temporal reasoning and generalization. Wu et al. [58] focus on short atomic action clips for situated reasoning. Yi et al. [44] generates synthetic videos for studying counterfactual predictions on collisions. Zadeh et al. [47] collects questions for social intelligence evaluation. Nevertheless, none of these benchmarks addressed the aforementioned critical dimensions of goal-oriented activity understanding from a real-world egocentric perspective.

# 3  The EgoTaskQA Benchmark

The EgoTaskQA benchmark contains 40K balanced question-answer pairs selected from 368K programmatically generated questions generated over 2K egocentric videos. We target the crucial dimensions for understanding goal-oriented human tasks, including action effects and dependencies, intent and goals, and multi-agent belief modeling. We further evaluate models' capabilities to describe, explain, anticipate, and make counterfactual predictions about goal-oriented events. A detailed comparison between EgoTaskQA and existing benchmarks is shown in Tab. 1.

## 3.1  Data Collection

We select egocentric videos from the LEMMA dataset [11] as base video sources. Compared to similar egocentric datasets, human activities in LEMMA are highly goal-oriented and multi-tasked. These activities contain rich human-object interactions and action dependencies in both single-agent and two-agent collaboration scenarios. We take advantage of these desired characteristics and augment LEMMA with ground truths of object states, relationships, and agents' beliefs about others. More specifically, we augment LEMMA on the following aspects:

**World States**   We focus on world states consisting of object states, object-object relationships, and human-object relationships. First, we build the vocabulary of relationships and state attributes from activity knowledge defined in previous works [37, 96]. We manually filter irrelevant relationships and attributes by removing dataset-specific (*e.g.*, under the car) and detailed numerical (*e.g.*, cut in three) relationships. Next, we gather similar relationships to obtain 48 relationships and 14 object attributes. This vocabulary covers spatial relationships (*e.g.*, on top of), object affordances (*e.g.*, openable), and time-varying attributes (*e.g.*, shape). We build on top of action annotations from LEMMA and use Amazon Mechanical Turk (AMT) to annotate this information before and after the changing action for all time-varying objects. With these annotations, we reconstruct the transition chain for each interacted object and obtain their temporal status. We provide the complete list of relationships and object attributes in the *supplementary*.

**Multi-agent Relationships**   To capture how two agents (actor and helper) collaborate over the same task, we annotate basic information about objects' visibility and the actor's awareness of the helper. For each object that the actor operates on, we annotate its visibility to the helper by providing synchronized videos from both agents' views to AMT workers. For the actor's awareness of others,

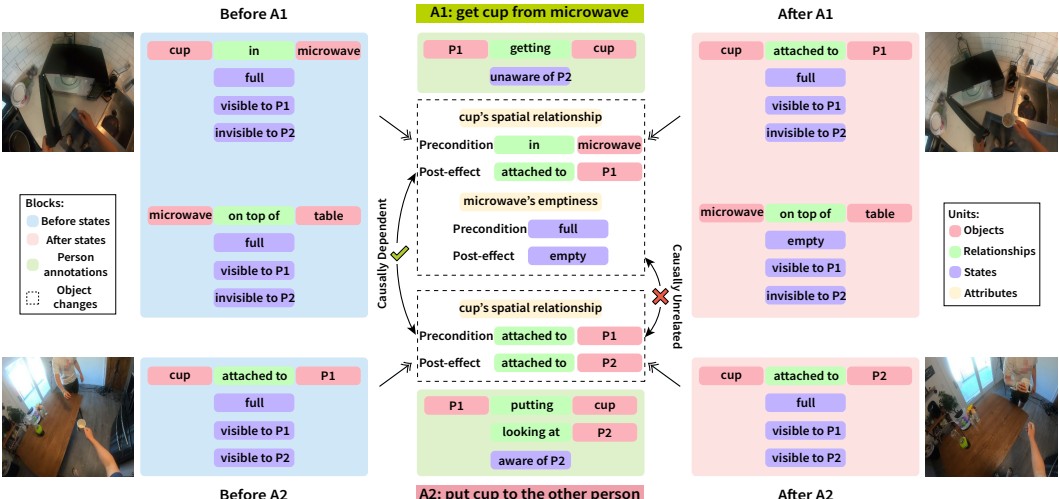

Figure 2: We use two actions A1:"get cup from microwave" and A2:"put cup to the other person" from Person 1's video in Fig. 1 as an example to visualize annotations in EgoTaskQA. We annotate states and relationships for objects changed by actions as well as human-object and multi-agent relationships. After obtaining the "before" and "after" annotations, we examine which attributes of objects were changed by the action and what are the preconditions and post-effects. We determine the causal dependency between actions by checking if there exists an object that the post-effect of one action over this object fulfills the preconditions of another. In this case, the state change of "cup" determines that A1 and A2 are causally dependent (best viewed in color).

we instruct AMT workers to first go through the egocentric view video of both agents to get familiar with actions performed by the actor and the helper. Next, we ask AMT workers to replay the video of the actor and annotate, during each action segment, whether the actor can see the helper or whether the actor is aware of the helper's action if the helper is not in sight. As this annotation is usually subjective, we take the majority vote of three workers as ground truth.

**Causal Trace**    Based on the annotated transition chain of objects, we generate causal traces for each action with rules. By checking whether the post-effect of one action fulfills the preconditions of another, we define the causal relationship between two actions into unrelated, related, and causally dependent; see Fig. 2 for an illustration and refer to *supplementary* for detailed explanations. Given a video, we run this dependency check for each pair of actions. Next, we generate a video-level dependency tree by recursively checking sequential depending relationships and use it as the ground truth dependency structure for subsequent explanatory and counterfactual question generation.

In total, we augment LEMMA with 30K annotated before states, after states, and person annotation blocks as shown in Fig. 2. We then segment the videos in LEMMA into clips with lengths of around 25 seconds for question generation. This design helps generate interesting clips with partially observed environmental constraints (*e.g.*, the cup is already washed when the person pours juice), and visual hints for future actions (*e.g.*, cutting watermelon into dice instead of pieces for making juice rather than eating it directly). Meanwhile, we keep our videos reasonably long, with an average of 5 actions per clip to cover sufficient information for action dependency inference and future prediction. We provide more details about data collection and annotation statistics in *supplementary*.

## 3.2    Question-Answer Generation

We use machine-generated questions to evaluate models' task understanding capabilities. We focus on the transition chain of each interacted object, especially what actions caused changes on objects and how these changes contribute together to a multi-step task; see examples in Fig. 1.

**Question Design**    We design questions that pinpoint scopes, including (1) action preconditions, post-effects, and their dependencies, (2) agents' intents and goals, and (3) agents' beliefs about others. Similar to [44], we categorize our questions over these three scopes into four types to systematically test models' capabilities over spatial, temporal, and causal domains of task understanding:

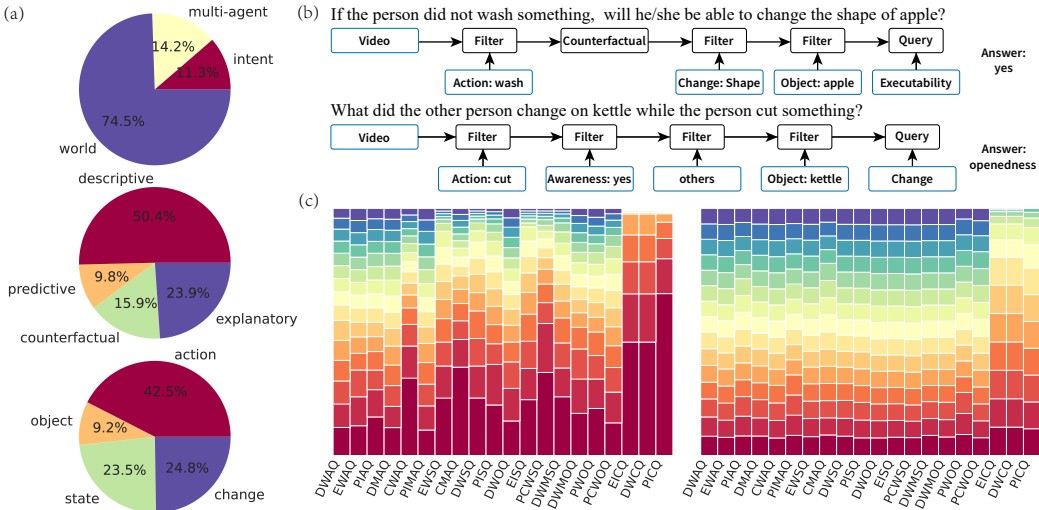

Figure 3: Generation and statistics of the question-answer pairs. (a) Question distribution according to question scope (top), question type (middle), and targeting semantics (bottom). (b) Examples of natural language questions and their corresponding executable programs. Operators and parameters of the program are represented by black and blue contour rectangles, respectively. (c) Answer distribution for the top 20 reasoning types and top 15 answers for each type before (left) and after (right) balancing, the reasoning types are abbreviated with the concatenation of their initial letters (*e.g.*, DWAQ for descriptive, world, action, and query).

- **Descriptive** questions evaluate the understanding of detailed spatial-temporal information. We provide spatial-temporal references in the questions to identify a unique interval for answering queries on objects and actions. These properties include object states and changes, relationships, human actions, and multi-agent-related information. We generate this type of question by randomly sampling an interval in the video clip and then gathering all related annotations for question generation. Answers in this category are generated based on the interval annotation and contain both open-ended queries and statement verifications.
- **Predictive** questions aim at understanding intents and task planning. Given a video clip, we ask about possible future object states and actions for both the actor and the helper. These predictions include both direct predictions on actions and objects, as well as more challenging task-dependent predictions such as the executability of actions and the desired states of objects. Questions and answers for predictive questions are generated by gathering the future action/object annotations in a fixed window size after the truncated interval (*i.e.* unseen future video) in the long original video. Answers in this category are open-ended action, object, and state queries.
- **Counterfactual** questions aim at understanding action preconditions and post-effects. Based on the causal trace of actions, we generate counterfactual questions with hypothetical conditions that certain actions in the sequence were not executed. Under this condition, we query both the affected and unaffected actions about their executability and whether the corresponding changes of object states associated with these actions will occur. We generate counterfactual questions by adding or removing actions in the causal trace and adjusting the depending actions' executability recursively. Answers in this category contain action executability verifications and object state queries.
- **Explanatory** questions evaluate the understanding of task-related object changes as well as action preconditions and post-effects. Given the object state annotations and the causal trace, we query the cause of state changes, the leading factor that satisfies the preconditions of specific actions, as well as why would the post-effect of certain actions affects other actions in the video clip. We generate explanatory questions by querying both the annotations as well as the causal trace. Answers for explanatory questions contain both open-ended and verification queries.

**Answer Generation** In EgoTaskQA, we consider both open-answer queries and binary statement verifications. To generate question-answer pairs for these questions, we design both text templates and the corresponding functional program templates as shown in Fig. 3 (b). Each program consists of a sequence of modules for querying the answer from the annotations and the causal trace. We exhaustively execute all possible program instantiations on videos to obtain answers by substituting arguments with instances in the available sample space. As all questions take action grounding as a

prerequisite, we add indirect references (*e.g.*, the first..., the action before...) to actions and objects when making substitutions to reflect this challenge; see details in *supplementary*. After this initial process, we obtain 368K question-answer pairs over 2K videos as the full question set.

**Answer Distribution Balancing**    We balance our answer distribution to avoid shortcuts from exploiting imbalances. Following the scheme introduced in [56], we tag each question template with its scope, type, and the targeting semantic category (*e.g.*, actions, objects, states) and use the composition of all tags as the unique reasoning type for each question. We balance binary verification questions to have an equal proportion of each answer within each reasoning type. For open-answer questions, we use rejection sampling to ensure that the top 20% frequent answers for each reasoning type do not appear as answers for more than 33.3% of questions in the same type. After balancing by reasoning types, we proportionally sample questions to obtain a 40K diverse and balanced question set with a 1:2 ratio of binary and open-answer questions. We visualize the statistics of questions and answers and the effect of answer balancing in Fig. 3. More details are provided in *supplementary*.

**Benchmark Splits**    We provide two benchmarking splits *normal* and *indirect* for video question-answering on EgoTaskQA. For the *normal* split, we randomly sample questions according to their answer distribution and reasoning types to have a 3:1:1 split over training, validation, and test sets. The *indirect* split is motivated by the fact that during task execution, actions, objects, and their changes are often strongly correlated. It leaves the chance for the model to perform well by simply over-fitting these strong correlations without thorough task understanding; see Sec. 4.2 for a more in-depth discussion. We leverage the indirect references in our question to inspect the models' capability to use the learned knowledge for multi-step reasoning and generalize them to indirect references without over-fitting. More specifically, we filter questions without indirect references and simple indirect reference questions without multiple reasoning steps (*e.g.*, what is the first action this person did? what did the person do before action "putting something"?) from all question-answer pairs to form the training set, and split all indirect reference questions with multiple reasoning steps as validation and test sets. Under this setting, the *indirect* split has a portion of 2:1:1 for training, validation, and test sets, respectively. We leave the remaining discussion of the *indirect* split to Sec. 4.3.

# 4    Experiments

In this section, we evaluate and analyze the performance of video question-answering models on EgoTaskQA. We report how well models perform on different question scopes, types as well as targeting semantics on both *normal* and *indirect* splits. We also provide diagnostic experiments on the language modality to show the necessity of the *indirect* split.

**Baselines**    In our experiments, we evaluate six state-of-the-art video question-answering models: VisualBERT [97], PSAC [98], HME [99], HGA [100], HCRN [101], and ClipBERT [102]. VisualBERT is a VL-BERT model designed for vision-language tasks. PSAC uses positional self-attention and co-attention network blocks to fuse visual and language features. HME uses external memory blocks for both visual inputs and questions on top of an LSTM-based encoder-decoder structure. HGA formulates video question-answering by constructing graphs for both videos and questions and aligning them. HCRN adopts a hierarchical framework by stacking relational modules over motion, question, and visual features. ClipBERT leverages sparsely sampled video clips and grid features [103] in a transformer architecture and achieves state-of-the-art results on video question-answering. We formulate question-answering in EgoTaskQA as a classification problem over all answer vocabulary and use models' accuracy as the evaluation metric under different settings. We provide details on model implementation, hyperparameter selection, and the training process in *supplementary*.

## 4.1    Comparative Analysis

We provide experimental results of baseline models on the EgoTaskQA *normal* split in Tab. 2. Model performances are evaluated on question scopes, types, targeting semantics, and overall answer categories. To quantify the naturalness and correctness of questions and answers in the EgoTaskQA benchmark, we provide human evaluation following the consistency check introduced in [87, 56]. More specifically, we randomly sample 50 questions for each category and instruct AMT workers to evaluate the quality of the generated answer. Additionally, we compare all baseline models with a simple frequency-based baseline, namely "Most Likely" in Tab. 2, where we select the most likely answer for each category to answer all questions in that category.

Table 2: Model performance on the EgoTaskQA *normal* split.

| | Category | Most Likely | VisualBERT [97] | PSAC [98] | HME [99] | HGA [100] | HCRN [101] | ClipBERT [102] | Human |
|---|---|---|---|---|---|---|---|---|---|
| **Scope** | world | 18.62 | 39.73 | 40.76 | 41.91 | 38.82 | **44.27** | 42.15 | 74 |
| | intent | 2.54 | 44.51 | 46.19 | 48.92 | 42.12 | **49.77** | 40.94 | 82 |
| | multi-agent | 10.92 | 26.29 | 30.59 | 27.98 | 23.43 | **31.36** | 27.63 | 76 |
| **Type** | descriptive | 18.64 | 41.99 | 40.63 | 41.45 | 38.04 | **43.48** | 38.45 | 88 |
| | predictive | 1.57 | 30.37 | 31.98 | 35.88 | 25.57 | **36.56** | 31.50 | 88 |
| | counterfactual | 23.62 | 41.99 | 41.89 | 44.13 | 41.94 | **48.00** | 46.75 | 80 |
| | explanatory | 7.97 | 37.42 | 37.99 | 38.85 | 35.97 | 40.60 | **42.39** | 74 |
| **Semantic** | action | 10.05 | 15.02 | 14.75 | 14.99 | 15.08 | 14.92 | **22.91** | 70 |
| | object | 2.07 | 23.26 | 36.53 | 36.05 | 19.09 | **45.31** | 21.80 | 82 |
| | state | 6.05 | 59.20 | 61.89 | 63.44 | 55.65 | **68.28** | 54.36 | 80 |
| | change | 41.97 | 68.27 | 65.05 | **68.87** | 68.38 | 67.38 | 66.58 | 82 |
| **Overall** | open | 0.70 | 24.62 | 26.97 | 27.66 | 22.75 | **30.23** | 27.70 | 82 |
| | binary | 50.46 | 68.08 | 65.95 | 68.6 | 68.53 | **69.42** | 67.52 | 76 |
| | all | 15.4 | 37.93 | 38.90 | 40.16 | 36.77 | **42.20** | 39.87 | 80 |

As shown in Tab. 2, the low performance of the most likely answer proves that our answer distribution is correctly balanced. For certain categories (*e.g.*, change), the most likely answer has relatively high accuracy (41.97%) as it covers both open-answer and binary questions. Next, we observe relative low human performance in certain categories (*e.g.*, action and explanatory). This indicates that identifying causal dependency between actions and conducting multi-step reasoning is not a trivial task for humans as also discovered in [56]. However, we still observe a large gap between state-of-the-art models and human performance. Among all models, we find HME, HCRN, and ClipBERT to perform the best. This result is reasonable since they leverage different ways to provide better visual representations and interactions between video and language. Among all question scopes, we recognize a relatively low accuracy on multi-agent-related questions among all question scopes. It implies that understanding other agents' actions during task execution is still difficult without explicit modeling. It is significant in egocentric vision as a person's view changes dramatically, and only glances can be taken to acquire others' information. Meanwhile, we notice that these models perform relatively well for questions on states and changing attributes. We conjecture that this is attributed to the task knowledge embedded in textual descriptions of questions since actions, objects, and state changes are strongly correlated, as mentioned in Sec. 3.2.

## 4.2 The Effectiveness of Language

**Object information**    We found the object information in the texts to be highly beneficial for question-answering on task-related knowledge during initial experiments. Compared to the original LEMMA action annotation (*e.g.*, drinking [cereal] with [cup]), we use verbs to refer to actions in EgoTaskQA and obfuscate object information at different levels (*e.g.*, drink something with cup, drink something with something) as similarly done in [56, 58]. While both types of action references localize to the same action interval, it contains different levels of knowledge in the language modality. Intuitively, the combination of action verbs (*e.g.*, cut) and targeting objects (*e.g.*, watermelon) provide object state information (*e.g.*, diced) under certain scenarios. Therefore, we compare models' performance at different levels of object information obfuscation. As shown in Fig. 4, we recognize a significant performance gain for all models by gradually removing object information obfuscation in text, *i.e.*, substituting "something" with the original object. This result supports the hypothesis that with fine-grained action annotations, we can learn task-related knowledge reasonably well by simply exploiting texts. It shares the same conclusion with recent works on leveraging text-based knowledge for helping instructional video understanding [104]. To further investigate the effectiveness of the language modality, we conduct ablative experiments on the EgoTaskQA *normal* split.

**Language-Only**    Language has been shown to provide knowledge that helps visual question-answering [105]. To study the role of language in EgoTaskQA, we design a text-only setting for VisualBERT and HCRN, testing BERT [106] and HCRN without vision against their vision-language counterparts. As shown in Tab. 3, the performance for most question categories dropped significantly. For the task of video question-answering, we should expect that dropping the vision branch will significantly affect the models' performance. As shown in Tab. 3, we observe the general performance for the two models decreased as we expected. Among all categories, the models' performance for the objects decreased the most, which is consistent with the fact that the object queries highly depend on the situation provided in the videos (*e.g.*, which object changed its status in the video?). However, we

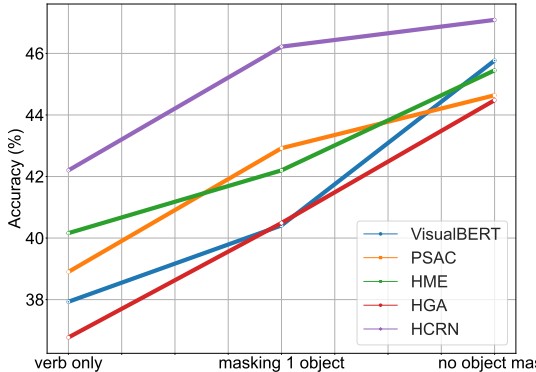

Figure 4: Ablative study on model performance with different levels of object information obfuscation on the Ego-TaskQA *normal* split.

Table 3: Language-only question-answering results on the EgoTaskQA *normal* split.

| Category | BERT [106] | | HCRN (w/o vision) | |
|---|---|---|---|---|
| | Acc. | Change | Acc. | Change |
| world | 36.28 | -8.7% | 35.22 | -20.4% |
| intent | 35.02 | -21.3% | 34.93 | -29.8% |
| multi-agent | 20.58 | -21.7% | 19.17 | -38.9% |
| descriptive | 34.55 | -17.7% | 33.58 | -22.8% |
| predictive | 24.75 | -18.5% | 24.3 | -33.5% |
| counterfactual | 41.3 | -1.6% | 40.4 | -15.8% |
| explanatory | 31.78 | -15.1% | 30.57 | -24.7% |
| action | 15.72 | +4.6% | 15.64 | -1.7% |
| object | 7.43 | -68% | 6.33 | -86.0% |
| state | 45.03 | -23.9% | 42.51 | -37.7% |
| change | 69.87 | +2.3% | 68.77 | +2.1% |
| all | 33.92 | -10.6% | 32.51 | -23.0% |

Table 4: Model performance on the EgoTaskQA *indirect* split.

| | Category | BERT | HCRN (w/o vision) | VisualBERT | PSAC | HME | HGA | HCRN | ClipBERT |
|---|---|---|---|---|---|---|---|---|---|
| Scope | world | 34.96 | 33.61 | 40.00 | **44.74** | 35.91 | 31.29 | 44.04 | 26.51 |
| | intent | 23.56 | 23.98 | 36.02 | **48.38** | 31.73 | 20.42 | 47.02 | 14.66 |
| | multi-agent | 19.70 | 19.25 | 26.02 | **35.37** | 25.07 | 17.74 | 30.11 | 20.09 |
| Type | descriptive | 33.09 | 30.73 | 38.9 | **43.36** | 34.48 | 29.01 | 42.02 | 24.35 |
| | predictive | 15.58 | 13.68 | 31.37 | 29.11 | 27.79 | 15.16 | **46.32** | 10.32 |
| | counterfactual | 34.59 | 34.75 | 37.63 | 39.94 | 35.07 | 33.01 | **43.64** | 26.29 |
| | explanatory | 27.38 | 28.11 | 32.75 | **42.53** | 29.16 | 24.00 | 39.69 | 22.46 |
| Semantic | action | 26.91 | 28.18 | 27.49 | **30.06** | 25.12 | 26.15 | 29.61 | 25.25 |
| | object | 2.808 | 4.13 | 22.63 | 30.97 | 19.08 | 7.02 | **32.20** | 10.49 |
| | state | 21.96 | 21.24 | 32.02 | **43.29** | 31.60 | 17.67 | 41.81 | 15.29 |
| | change | 55.28 | 50.71 | 55.59 | **57.20** | 47.65 | 47.22 | 56.27 | 35.26 |
| Overall | open | 11.22 | 11.38 | 21.05 | **28.23** | 18.27 | 8.66 | 27.82 | 11.17 |
| | binary | 58.24 | 55.52 | 57.61 | **60.30** | 52.55 | 53.72 | 59.29 | 40.71 |
| | all | 31.78 | 30.76 | 37.01 | **42.25** | 33.06 | 28.36 | 41.56 | 24.08 |
| Performance Change | | -6.4% | -5.4% | -2.4% | +4.9% | -17.7% | -22.9% | -1.5% | -39.6% |

observe a slight performance gain on object state change questions. This further suggests that the knowledge of world state change, *i.e.* which object attribute could change under actions, is embedded within question texts. Models could exploit question texts to learn simple associations between attribute types and action verbs (*e.g.*, cleanliness and wash, emptiness and pour, shape and cut, *etc.*).

## 4.3 Generalizing to indirect references

On the EgoTaskQA *indirect* split, we evaluate models' capability to leverage learned task knowledge for solving more complicated indirect reference tasks. With the *normal* split allowing for shortcuts on action-state associations, the *indirect* split forbids such exploitation by differentiating references during training and testing. As shown in Tab. 4, we observe more significant performance drops in language-only models compared to their vision-language counterparts. More specifically, the performance of BERT and language-only HCRN dropped 20.8% and 26.3% on the "change" category, where we observed potential exploitation on question texts in Sec. 4.2. This serves as a shred of evidence that the *indirect* split helps reduce the possibility of exploiting simple associations in texts. As for baseline models, we recognize a common performance decrease shared by most models on the *indirect* split. Among them, we notice a significant performance drop for ClipBERT, which conflicts with the dominating role of large-scale pretrained vision-language models on various reasoning tasks. We suspect that this degeneration might originate from two lines of problems: (1) the model design on sampling fewer videos and aligning visual/text graphs directly, which conflicts with the intuition that detailed spatial-temporal information and reasoning is indispensable for grounding indirect references; and (2) adopting large-scale pre-trained models directly to a specific domain is non-trivial, especially with challenges in grounding knowledge to visual signals. Overall, our experiments on the

EgoTaskQA *indirect* split further reveals the demand for better spatial-temporal reasoning modules that solve the problem of compositional goal-oriented reasoning with indirect references.

# 5 Conclusions

We introduce the EgoTaskQA benchmark to systematically evaluate models' understanding of goal-oriented activities from an egocentric perspective. We annotate object states, relationships, and agents' beliefs on the LEMMA dataset. We generate diverse questions covering different reasoning capabilities and target the crucial dimensions of task understanding: action dependencies and effects, agents' intents and goals, and belief modeling. We evaluate state-of-the-art video question-answering models and show their gaps compared with the human on two challenging splits, *normal* and *indirect*, to promote future study on indirect reference understanding and goal-oriented reasoning.

**Ethics** The EgoTaskQA benchmark is built upon LEMMA and contains different subjects. As noted in LEMMA, the authors obtain consent from subjects by signing an agreement form with potential impacts adequately informed before recording. We mainly annotated objective world status and multi-agent information using multiple-choice selection and asked annotators to annotate the awareness of other subjects' actions with binary options for subjective annotations. For the annotation process, the workers' agreements are obtained by the publicly available annotation service platform AMT we adopted. In summary, we forbid subjective comments with no personally identifiable information revealed, and all participants' agreements are well addressed.

**Limitations** Our work is primarily limited to two aspects. (i) Constrained by the data and annotation complexity, the scope of our activities is limited to indoor goal-oriented tasks. We believe that adding more diverse activities to EgoTaskQA will further increase its value as a general benchmark. (ii) Although EgoTaskQA supports many more additional supervisions, we currently limit our discussions to the six state-of-the-art models to have a fair comparison. With the current result showing insights into problems and challenges in EgoTaskQA, we leave the exploration of model design to future work and briefly discuss the potential solutions as follows. Meanwhile, we will continue this data curation for a broader range of human activities.

**Future work** We plan to investigate the following two branches in the future, (i) explicit spatial-temporal grounding for modularized video QA models and (ii) prompting large-scale pre-trained models (both visual and language) for the domain-specific video QA challenges. Firstly, egocentric data can provide finer information and ease the challenge of grounding in modularized neuro-symbolic models. This could complement existing video reasoning methods and test the potential of neuro-symbolic models on complex reasoning tasks from a real-world, multi-agent, and causal perspective. Next, with increasing efforts in adapting large-scale pre-trained models for reasoning, our experiments suggest that adopting such models directly to a specific domain is non-trivial. Compared to their capabilities in commonsense reasoning, how to enable pre-trained models with the ability to fastly adapt to complex reasoning tasks still remains an interesting problem to be solved.

**Broader impact** With most existing intelligent robots depending on the understanding of world states to act and plan, we hope that the augmented LEMMA dataset can bridge the study of world-model learning in simulated environments and real-world complex event understanding. Additionally, we believe the EgoTaskQA benchmark proposes challenges on goal-oriented reasoning and hope such efforts can foster research in broader video understanding directions, including video-language understanding, spatial-temporal grounding, task learning, future anticipation. We also see its potential in imitation learning and knowledge acquisition, which will further drive the study of intelligent assistive robots that can perform tasks coordinately with humans.

**Public access** We host our videos, annotations, metadata, and question-answering pairs on our website. We provide videos in .mp4 format, metadata, annotations in JSON and pandas DataFrames, and question-answer pairs with their corresponding metadata in JSON format. We make our data publicly available under the CC BY-NC-SA license, which allows reusers to distribute, remix, adapt and build upon the material for noncommercial purposes only and only so long as attribution is given to the creator. We bear all responsibility in case of violation of rights.

**Acknowledgement** We thank all colleagues from VCLA and BIGAI for fruitful discussions. We would also like to thank the anonymous reviewers for their constructive feedback. This work reported herein was supported by National Key R&D Program of China (2021ZD0150200).

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
