# OpenReview forum: "EgoTaskQA: Understanding Human Tasks in Egocentric Videos"
_NeurIPS.cc/2022/Track/Datasets_and_Benchmarks — NeurIPS 2022 Datasets and Benchmarks _

### Official Review · Reviewer_iVFc · 2022-07-21
**Solid new egocentric QA benchmark but no option to review the actual dataset**

**Rating:** 5
**Confidence:** 4
**Clarity:** The paper is generally well-written a…

**Strengths:**

- The work significantly expands an existing egocentric dataset with 40K questions covering 4 different types. This dataset will be very useful to the embodied AI and VQA/VQC research communities.
- The paper comes with a comprehensive related work with a summary table that contrasts the contributions of this work w.r.t. relevant literature.
- Evaluation compares 6 VQA models. Several ablation studies are also present to demonstrate the usefulness of object information and language supervision (albeit masking with a common term vs. using the noun terms does not necessarily imply better action understanding and further analysis is needed)

**Weaknesses:**

- Unfortunately, reviewers are not able to access the data without exposing their identities. The website mentions filling in a form/license data agreement for downloading the data with a note:  *During review process, we refer to the website for data examples and temporarily forbids full data download.* It is unclear how the dataset will be distributed afterward, i.e., i) on which platform (the datasheet in the supplementary material mentions *dataset could be accessed on our website* but would that be restricted access?) and ii) whether all code will be released and scripts for ease of reproducibility of the reported experiments will become available.

- It is unclear how the correctness of answers generated by functional programs is verified. Similarly, how questions are machine-generated remains fairly unexplained. To my understanding, neither the code for the QA construction is not open-sourced nor the paper contains sufficient details on QA quality. Is the evaluation of the quality of the generated answers limited to the one described in lines 239-241 (randomly sampling 50 questions from each category)? It seems that some categories have very low accuracy.

- The observed performance increase of text-only models on object state change questions is worthy of further analysis. Perhaps the associations captured are not action-aware or context-aware, but rather simplistic linguistic co-occurrence patterns, as the experiments in Section 4.3 suggest. However, the performance drop between normal and indirect splits for text-only models is marginal. Would be nice to open source models, both for reproducibility and to allow future research to perform more exploration on what exactly language-guided models are able to learn.

**Additional Feedback:**

- Have the authors considered an object-centric split to evaluate how models generalize to unseen objects?

**Correctness:**

It is difficult to judge correctness with a few examples presented in the paper and website. Most importantly, the paper does not contain a lot of information on whether it is safe to assume that template-based QA generation is accurate, or whether any additional verification has taken place.

Other minor comments:
- Please further explain how spurious correlations are avoided by including direct and indirect references to actions and objects (lines 51-52)
- Please explain lines 130-132, specifically add a brief description of the vocabulary of relationships and state attributes and how manually filtering of irrelevant relationships is performed.
- What is the inter-annotator agreement for AMT workers on tasks described in the Multi-agent Relationships section?


**Documentation:**

Insufficient details. General statements and a data sheet are provided but there is no URL for reviewer access to the dataset.


**Relation To Prior Work:**

There is some closely related work missing, in particular, similar datasets in embodied AI frameworks [1,2] and models that learn world dynamics (object state changes and object interactions) [3].

[1] Padmakumar, Aishwarya, Jesse Thomason, Ayush Shrivastava, Patrick Lange, Anjali Narayan-Chen, Spandana Gella, Robinson Piramuthu, Gokhan Tur, and Dilek Hakkani-Tur. "Teach: Task-driven embodied agents that chat." In Proceedings of the AAAI Conference on Artificial Intelligence, vol. 36, no. 2, pp. 2017-2025. 2022.

[2] Gao, Difei, Ruiping Wang, Ziyi Bai, and Xilin Chen. "Env-QA: A Video Question Answering Benchmark for Comprehensive Understanding of Dynamic Environments." In Proceedings of the IEEE/CVF International Conference on Computer Vision, pp. 1675-1685. 2021.

[3] Zellers, Rowan, Ari Holtzman, Matthew Peters, Roozbeh Mottaghi, Aniruddha Kembhavi, Ali Farhadi, and Yejin Choi. "PIGLeT: Language grounding through neuro-symbolic interaction in a 3D world." arXiv preprint arXiv:2106.00188 (2021).

**Summary And Contributions:**

This work introduces a new video question answering benchmark that consists of egocentric videos with fine-grained annotations of object states, object-object, human-object and multi-agent relations, and causal action relations. The dataset also contains four types of question-answer pairs, including counterfactual and explanatory questions, in addition to questions that aim to capture intents, goals and object states and changes. Finally, the work compares 6 video question answering models on two data splits.

---

> ### Author Response · Authors · 2022-08-18
> **Response to the reviewer (3/3)**
>
> ### 5. Line 130-132, add a brief description of the vocabulary of relationships and state attributes and how manually filtering of irrelevant relationships is performed.
> We refer the reviewer to the general responses for the vocabulary of relationships and state attributes (the statistics can also be visualized in the supplementary). As we discovered, the gathered state/object collection from existing activity knowledge bases contained many object-related and detailed numerical relationships (e.g. in measuring cup, cut in three, in foil, etc.), we remove irrelevant object-related relationships by checking if the objects in these relationships are annotated in the original LEMMA dataset. We also remove all specific numerical relationships and use a coarser one that exists in the collection for substitution (in three -> diced, in part). Following a similar principle, we group object states and relationships by their meaning and define representative attributes with available values for the ease of annotation (e.g., "shape" for aggregating ["diced", "in parts", "a whole"]). We will add this description in the corresponding place in the revision.
>
> ### 6. What is the inter-annotator agreement for AMT workers on tasks described in the multi-agent relationships section?
> During the annotation process, we independently instruct each annotator to watch, replay and identify the visibility/awareness of others or objects (as illustrated in the "Multi-agent Relationships" paragraph in Sec.3.1). We do not add additional instructions or provide no further examples on what are the scenarios for visible/invisible or aware/unaware. Therefore, as stated in Sec.3.1, this annotation is purely subjective depending on how the annotator believes what the video subjects think and we only take the majority vote of three viewers as ground truth annotation.
>
> ### 7. Missing of related work, "Env-QA", "PIGLeT", "Teach: Task-driven embodied agents that chat"
> We thank the reviewer for the pointer and will add the following discussion in the main text: (1) we have already included the Env-QA in the original text (Tab.1 and L.103-104), and we will add an additional sentence to address its similarity to ours in the "Video Question Answering Benchmarks" paragraph by discussing the shared ingredients of egocentric view (from a robot's view) and object status; (2) we will add the discussion for PIGLeT and TEACh in the "Action as Inverse Planning" paragraph by discussing the connection of language grounding (i.e., action) to world state transitions as an emerging field for studying embodied planning and dialogue agents.
>
> ### 8. Have the authors considered an object-centric split to evaluate how models generalize to unseen objects?
> This is indeed an interesting setting that we considered at the initial stage of this project. In fact, directly reasoning over unseen objects is usually dependent on task understanding since objects and tasks are strongly correlated (i.e., unseen objects settings oftentimes indicate unseen tasks and thus are inappropriate for directly estimating task understanding capabilities). Therefore, we thought of designing a compositional generalization split for novel action-object compositions (i.e., using different verb+object to have a novel task execution order for a known task). However, such compositions are majorly limited by the variety of multi-step activities (tasks) in the dataset and are not available to serve as a balanced split in our benchmark given that our base video source, LEMMA, mainly covered a rather limited domain of activities. We believe such studies will definitely be an interesting topic on larger-scale video sources with a significantly larger task variety. Limited to the data and annotation resources we have, this branch of study is out of the scope of this project and is left to future work.

---

> ### Author Response · Authors · 2022-08-18
> **Response to the reviewer (2/3)**
>
> ### 4. Performance increase of the text-only models, marginal performance drop for normal and indirect splits, using the noun term does not necessarily imply better action understanding, how spurious correlations are avoided.
> We thank the reviewer for the insightful comments on linguistic perspective. Indeed, our original intuition for designing the object masking scheme and indirect split originated from the concern of spurious correlations in the language domain. In fact, we can easily think of several actions and state relationships that can be simply concluded from the text, e.g., "open <u>milk</u>" indicates "milk" is "opened". This originates from the semantics of action verbs since many verbs (e.g., open, close, cook, etc.) already contain sufficient knowledge for affordance and state change.
> To avoid such correlations, we first mask the objects in the compositional action annotation in LEMMA, i.e., from "open <u>milk</u>" to "open <u>something</u>", to remove the shortcut for object grounding from language. The model thus has to understand the verb, align it with the potentially multiple occurrences in a video segment, and use information in the text to ground the targeting object in order to solve the question. Next, to avoid shortcuts from semantically meaningful verbs (e.g., asking the status of "milk" with a verb "open" will likely lead to the status “opened”), we instead use the previous or the next action as a pivot to index the action verb. The question thus contains no direct correlation between action verbs and the answer since the verb is referenced by other actions. We did not consider more subtle correlations in the action sequence (e.g., after one video subject opens something, he/she will generally put something into it and then close it) in the current version as we figured a one-step indirect reference is already challenging enough for models.
> To verify such points with experimental results, we refer the reviewer to Fig.4, Tab.2, and Tab.4 in the main text. First of all, we can see from Fig.4 that removing the object mask will generally lead to a performance increase for all models indicating that there exist shortcuts for exploiting object information in the question. Next, if we compare Tab.2 and Tab.4 on the "change" and "state" categories, we can observe that all models had a significant performance decrease, supporting our assumption that removing spurious correlation will partially forbid models' exploitation of the action-state change from the text. Although the overall performance decrease seems marginal (balanced by the performance increase on the "action" category since during training we do not provide any complicated reference understanding, thus providing cleaner data for understanding the action itself and its properties), the "indirect" split does support our original design motivation for testing models' capability without shortcuts in language domain for exploitation. We have provided the link to model checkpoints in the response to the google form on our website, the reviewer is more than welcome to apply for the weights with an anonymized email address.

---

> ### Author Response · Authors · 2022-08-18
> **Response to the reviewer (1/3)**
>
> We thank the reviewer for the insightful feedback, acknowledging the potential impact of our work on the embodied AI community, the clear contrast with previous works, and comprehensive experimental results.
>
> ### 1. The dataset is not available to access the data without exposing their identities. It is not clear how the dataset will be distributed (would the access be restricted? and whether all code will be released and scripts for ease of reproducibility of the reported experiments will become available)
> Please refer to the general responses for the dataset release and the question generation code release. We apologize for the unclear instructions for dataset access during the reviewing period as we had concerns on OpenReview and did not directly release it to the public (we can only make the data publicly available on the internet before, now we can reveal it only to the reviewers), and therefore provided as many examples as the website can afford to visualize. For dataset hosting, we plan to use the provided [link](https://sites.google.com/view/egotaskqa) and google drive for dataset storage. Each participant will be asked to fill out a form for knowing the license agreement, as well as stating their purposes for accessing the data and pre-trained model weights. The access will not be restricted, as long as the applicant acknowledges all licensing agreements. Again, we followed previous works on designing this data application pipeline (e.g. iGibson, LEMMA, Env-QA, etc.) and did not intend to acquire any reviewer information (using an anonymous email and name can download the checkpoints as long as the email address is valid).
>
> ### 2. How questions were machine-generated remains fairly unexplained
> We make a more detailed illustration of the question generation pipeline. First of all, we refer the reviewer to the design of templates and indirect references shown in Tab.4 and Tab.5 in the supplementary. As we can see from the templates, we have place-holder blanks in the template like "\<o\>", "\<t\>", "\<a\>", "\<f\>". These blanks in the natural language links to the corresponding branch in the question program "obj@\<o\>", "before|after", "action@\<a\>", etc. As we make each program executable on the world state annotations, with proper substitution of these templates (i.e., initialized), we are able to execute the programs to get an answer. To find all such substitutions or initialization of the templates, we enumerate all possible objects, actions, time-hint (before, after), and status that exists in a video, and exhaustively executes all initialized programs. Next, the program executor will return an error message if one initialized program is not executable or has no answer (i.e., not valid). In this way, we filter all invalid initialized programs and use their corresponding natural language to pair the answer to collect the 368K question-answer pair for post-processing on answer distribution balancing.
>
> ### 3. Quality and reproducibility of the question-answer generation process, low accuracy on certain categories.
> In this work, we followed the verification process discussed in AGQA and GQA (around 50-100 questions per-category) to verify the questions by instructing AMT workers for solving 50 questions sampled from each category (including different question scopes, question types, and answer semantics). We provide the answer vocabulary to each worker and make the verification process a multiple-choice problem. We have also released our code for question generation, specifically, the question templates and the generation pipeline. We hope this information can sufficiently support the validity of our generated questions. As for the low accuracy for certain categories, we argue that this suggests the task of identifying causal dependency between actions and doing multi-step reasoning for the rationales of actions is also a non-trivial task for humans. Additionally, the difficulty is further aggravated by the indirect reference and object-masking for actions under certain scenarios (as AMT workers' responses suggest) where specific questions target challenging reasoning capabilities (e.g. explanatory or counterfactual). This phenomenon is also reflected in AGQA where a solver has to first ground on complex references of spatial-temporal tubes for solving the following reasoning task (e.g. the logic category and action category in Tab.2 [1]). However, we argue that the large performance gap between model performance and human performance (compared to AGQA, e.g. 69.9 for HCRN and 70.69 for humans in Tab.2 [1]) still indicates a gap between models and humans for identifying causal dependency between actions and doing multi-step reasoning for the rationales of actions.
> [1] Grunde-McLaughlin, Madeleine, Ranjay Krishna, and Maneesh Agrawala. "AGQA: A benchmark for compositional spatio-temporal reasoning." CVPR 2021.

---

### Official Review · Reviewer_CuX8 · 2022-07-24
**Updated Review for EgoTaskQA**

**Rating:** 6
**Confidence:** 3

**Strengths:**

- Based on Table 1, and the authors review of related works, there is a richer set of questions in this dataset compared to baseline (although I have some concerns about the questions themselves, see first point in weaknesses). The proposed question types (descriptive, predictive, counterfactual, explanatory) and the generated causal dependency relationships are interesting for understanding the performance of Video QA models.

- The authors benchmark a set of state-of-the-art video models on their dataset, with both normal split as well as a split for studying indirect references. These benchmarking efforts helps the community understand the strengths & weaknesses of existing models.


**Weaknesses:**

- The full dataset is not available at this stage, even to the reviewers (please correct me if I'm wrong). Also, based on samples I'm seeing from the data, the automatically generated questions in this work seems a lot less clear than prior work such as AGQA [A]. From the author's website, some samples "Q: What does the person want watermelon to be for doing the action cut something using something in the video?" "Q: What will the status of the last object that has status change change to if the actor pour from something into something in the future?". Could the authors clarify why the generated questions here are less clear?

- Due to the above, I have concerns about the ability of this dataset to evaluate QA of the different question types proposed by the authors. Furthermore, comparing Table 2 here with Table 2 in AGQA, many of the question categories are fairly similar. Since this dataset (40K questions) is also a lot smaller than AGQA (3.6M), the significance of this dataset is not clear to me - would appreciate clarifications from the authors.

- There was just 1 trial run for all the benchmarking. While I understand that running video models are expensive, this does bring some concerns on the variability of results across runs and reproducibility. The ML reproducibility framework was not used here.

[A] AGQA: A Benchmark for Compositional Spatio-Temporal Reasoning

**Additional Feedback:**

Could the authors let me know where reviewers can download the dataset? The link provided by the authors links to their public website, which currently requires a signed form for download and says "*During the review process, we only provide model checkpoints and temporarily forbid full data download".

**Clarity:**

The paper is generally well written, but I would appreciate more explanation on how the questions are generated in Section 3.2 (see my comments on weaknesses above).

**Correctness:**

- The full dataset is currently not available to reviewers (as far as I'm aware), but some samples are available on the website.

- The benchmark evaluations are present in the paper (I just have some concerns on reproducibility, see documentation section below).

* Post rebuttal - The dataset is presented to reviewers now and authors have added more comments on the documentation & reproducibility.

**Documentation:**

Some concerns I have on the documentation:

- The ML reproducibility checklist is not available for this paper and only one run is performed for the benchmarks.

- Of the date of this review, the data statistics page is empty (https://sites.google.com/view/egotaskqa/statistics).

- Reviewers don't have access to download the full dataset.

- There seems to be no details on the hosting plan (line 310 in supplementary).


**Ethics:**

Some items in the paper checklist are not available, notably 4 (d), 4 (e), and 5 (b) (see last page of the main paper).

Considering this is a dataset containing human subjects with annotations from Amazon Mechanical Turks, I'd like to check if this warrants an ethics review to make sure that the dataset doesn't need any IRB approvals / has sufficient consent from participants / does not contain offensive content. Additionally, the full dataset is not available to reviewers at this stage (as far as I'm aware).

**Relation To Prior Work:**

- The authors clearly discuss related works in many areas.

- I'd like more discussions from the authors on the significance of this benchmark with respect to AGQA (see weaknesses).

**Summary And Contributions:**

This dataset uses videos from the LEMMA dataset (egocentric videos of human-human / human-object interactions) for studying video QA. The authors extend LEMMA with annotations of objects, agents, and their relationships from Amazon Mechanical Turk. Then, the authors build causal dependency relationships between agents and objects in the videos. The questions in the dataset used in QA is then automatically generated with 4 types of questions (descriptive, predictive, counterfactual, explanatory). The dataset is evaluated on a set of 6 existing video models, showing a gap from the model to human performance.

---

> ### Author Response · Authors · 2022-08-18
> **Response to the reviewer (4/4)**
>
> ### 4. Concerns for ML reproducibility
> We provide the ML reproducibility checklist as follows:
> - A clear description of the mathematical setting, algorithm, and/or model
>     - Please check the "Baseline" paragraph in Sec.4 (main text) and all model description paragraphs in Sec. C (supplementary).
> - An analysis of the complexity (time, space, sample size) of any algorithm
>     - N/A, we adopted existing models with no direct adjustments on the model side.
> - A link to a downloadable source code, with specifications of all dependencies, including external libraries.
>     - Please check the GitHub URL we provided in the main text and supplementary.
> - Theoretical claims (a statement of results, explanations of assumptions, and proofs of the claim)
>     - N/A, this paper does not propose new algorithms
> - A complete description of the data collection process
>     - Please find the data collection details in Sec.3 in the main text as well as Sec. A and Sec. B in the supplementary.
> - A link to a downloadable version of the dataset
>     - Please refer to the general responses for dataset downloading.
> - An explanation of any data that were excluded, descriptions of any preprocessing-step
>     - Please refer to the data collection and question-answer balancing process in Sec.3 (main text) and Sec. A,B (supplementary) for details.
> - An explanation of how samples were allocated for training/validation/testing
>     - Please refer to the benchmark split paragraph in Sec.3.2 (main text).
> - The range of hyperparameters considered, methods to select the best hyper-parameter configuration, and specification of all hyper-parameters used to generate results
>     - Please refer to Sec. C (supplementary) for details.
> - A description of how experiments were run
>     - We train each model once on the training set of different dataset splits.
> - A clear definition of the specific measure or statistics used to report results
>     - We use the classification accuracy (in percentage) over all potential answers as the evaluation metric.
> - A description of the computing infrastructure used
>     - Please refer to Sec. C in the supplementary for details.
>
> As the reviewer mentioned, training video models from scratch is an extremely computation-heavy visual task that requires a tremendous amount of computing resources. Limited by our resources, we are only allowed to train each model once on each dataset to search for the best hyperparameters. Meanwhile, we notice that all existing models we selected in our baseline experiments report **single-run evaluation** results after a hyperparameter search in their original papers (please refer to their original papers for verification), which is treated as a common rule. As we mainly examined existing models without model-side adjustments, we believe this also indirectly supports that the models we selected are rather stable and that our experiments should be easily reproducible with the available training/hyper-parameter specifications.
>
> ### 5. Some items in the paper checklist are not available, 4(d) "how consent was obtained from people whose data you are using/curating", 4(e) "whether the data you are using contains personally identifiable information or offensive content", 5(b) "Did you describe any potential participant risks, with links to IRB approvals?", check if this warrants an ethic review to make sure that the dataset doesn't need any IRB approvals.
> For these items, we provide further clarifications for data collection details.
> * For 4(d), we adopted a publicly available annotation service platform Amazon Mechanical Turk. The agreement of workers for annotating the data is handled by the platform.
> * For the video data, we adopted a publicly available dataset LEMMA, therefore, we assume that all participants in the video have made the agreement when LEMMA is published.
> * For concerns in 4(d) on our additional annotations (as released in the dataset access, see general responses), we mainly annotate objective world and multi-agent information using multiple choice selection annotation on top of each action interval in LEMMA.
> * For the subjective annotations in our dataset, we only asked annotators to annotate whether a video subject can see the other and is aware of the other's actions, with no other information annotated. Answers to these two questions are also limited to "yes", and "no" during annotation.
>
> In summary, we provide no additional place for subjective (personal) comments and believe that such an annotation pipeline will not reveal any personally identifiable information or offensive content. For 5(b), we used a publicly available annotation procedure with no subjective opinion gathered and believe there are no potential participant risks.

---

> ### Author Response · Authors · 2022-08-18
> **Response to the reviewer (3/4)**
>
> ### 3. Similar to AGQA on question types and significantly smaller than AGQA, more discussion on the significance of this benchmark w.r.t. AGQA.
> As requested, we make further clarifications on our contribution and major differences with AGQA to solve the potential misunderstanding.
> 1. Goal-oriented understanding. The major goal of our work aims at a holistic understanding of tasks (multi-step events with explicit or implicit dependency between sub-events), including actions' preconditions and effects on objects, acting subjects' intent on how to accomplish the task, and how multi-agent collaborate together to finish the same task. This originates from the hope that an intelligent agent will be equipped with proper tools to understand and help humans by simply watching how human performs tasks (e.g., formulated as an RL problem in [1]). With these motivations, we selected LEMMA as our video source instead of Charades (AGQA's video source). Charades videos are mostly single-agent activities unrelated to any specific goal or task [2]. This gives the first major difference between our work and AGQA, where AGQA questions cover spatial-temporal reasoning questions, **but these questions are not goal-oriented**. Such differences are further revealed in our definition of world states, where ours consider task-relevant object attributes including states and affordances (as shown in Fig.2 in the supplementary) in addition to the spatial relationships considered in AGQA.
> 2. Various question types. We respectfully point out that our questions are different from AGQA ones. As far as we know, AGQA only covers questions with types "descriptive", "action" and "object" listed in our categories. If we look back to Fig.1's object-query example, we can see that the way that we deliver such "descriptive" and "object" questions in a task or state-oriented way, where such questions in AGQA mostly follow the form, "What did the person hold after putting a phone somewhere" (example from Fig.1 in [3]), still focusing on the spatial-temporal understanding of actions and objects instead of viewing from a task perspective.
> 3. Egocentric understanding. Our augmented-LEMMA dataset and EgoTaskQA are originally designed for evaluating understanding in egocentric videos. Compared to third-person videos (e.g. Charades videos used in AGQA), egocentric videos reveal new challenges in fine-grained activity understanding given their goal-oriented perspective and partial observability. As compared with existing works in egocentric vision (discussed in Sec.2), our benchmark can also serve as a systematic evaluation tool for fine-grained egocentric video understanding.
> In summary, we hope that the clarifications have cleared the reviewer's concerns on the novelty and significance of our work. We are also open to any further discussions with the reviewer on this point.
> [1] Puig, Xavier, et al. "Watch-and-help: A challenge for social perception and human-ai collaboration." ICLR 2021.
> [2] Jia, Baoxiong, et al. "Lemma: A multi-view dataset for learning multi-agent multi-task activities." ECCV 2020.
> [3] Grunde-McLaughlin, Madeleine, Ranjay Krishna, and Maneesh Agrawala. "AGQA: A benchmark for compositional spatio-temporal reasoning." CVPR 2021.

---

> ### Author Response · Authors · 2022-08-18
> **Response to the reviewer (2/4)**
>
> ### 2. Questions are less clear than prior work such as AGQA,  "Q: What does the person want watermelon to be for doing the action cut something using something in the video?" "Q: What will the status of the last object that has status change change to if the actor pours from something into something in the future?". More details on the question generation process.
> Here we provide clarifications for the mentioned examples and question design intuitions. As mentioned in the main text, our work augmented the LEMMA dataset with world state and multi-agent annotations. We kindly remind the reviewer that the original LEMMA dataset provides compositional action as annotations instead of simple verb-noun combinations. Therefore, the original action annotation in the mentioned questions are "cut <u>watermelon</u> with <u>knife</u>" and "pour from <u>basin</u> into <u>sink</u>".
>
> Next, if we get back to the two example questions, our full question without any information masking should be "what does the person want watermelon to be for doing the action cut <u>watermelon</u> using <u>knife</u> in the video?" and "what will the status of the last object that has status change change to if the actor pour from <u>basin</u> into <u>sink</u> in the future". These two questions are instantiated by two question templates "What does the person want [Object] to be for doing the action [Action]" and "What will the status of [Object] change to if the actor [Action]?". During the generation, we fill in the [Action] and [Object] blanks with available actions ("cut watermelon with knife", "pour from basin into sink") and objects ("watermelon", "basin"). Notice here that there exist indirect references in the second question (i.e., the object that has status change), which came from selecting unique attributes for describing the object (i.e., basin).
>
> The necessity of masking objects in questions: under these question generation settings, as mentioned in Sec.4.2, we discover that models actually exploit the fact of available object knowledge in the text for answering this question, e.g., "cut" + "watermelon" -> "diced". This is the key insight for the experimental results visualized in Figure 4., where we see a significant improvement of all models when feeding them unmasked action descriptions (though still inferior compared with the human performance in masked scenarios). As for the question that contains an indirect reference, we refer the reviewer to the video provided on the website. We can see that the reference "the last object that has status change" is the basin since the person gets it from the table. As a question for intent prediction, we know that he finished changing the water for the tank and will pour the water into the basin, then we can naturally predict that it will be "empty" in the future. However, if we do not mask the objects in the action annotation, we do not need such a reasoning process (i.e., identify the object -> conduct counterfactual reasoning on pouring something from it) and can directly get the answer "empty" from "pour from <u>basin</u> into <u>sink</u>" since the indirect reference to the object is clearly revealed in the action and the action itself contains enough state transition information (i.e., pour + basin -> empty basin).
>
> Above all, the aforementioned facts motivated us to provide a coarse (no direct correlation between answer and question texts), yet localizable (unique enough to serve as an index for localizing an interval or object in the video) indirect action/object description during the question-answering process.

---

> ### Author Response · Authors · 2022-08-18
> **Response to the reviewer (1/4)**
>
> We thank the reviewer for the insightful comments and the positive feedback on the variety of reasoning capabilities in question answering, that the causal dependency relationship is interesting, and the potential impact of the experimental results on the field. We address the reviewers’ concerns as follows:
>
> ### 1. The full dataset is not available at this stage, future dataset hosting plan, website flaws
> Please refer to the general responses for accessing the world-state/multi-agent annotations as well as the question-answering benchmark. We apologize for the unclear instructions for dataset access during the reviewing period as we had concerns on OpenReview and did not release it to the public (we can only make the data publicly available on the internet before, now we can reveal it only to the reviewers), and therefore provided as many examples as the website can afford to visualize. For dataset hosting, we plan to use the provided [link](https://sites.google.com/view/egotaskqa) and google drive for dataset storage. Each participant will be asked to fill out a form for acknowledging the license agreement, as well as stating their purposes for accessing the data and pre-trained model weights. For the empty link to the dataset statistics on our website, we have already fixed the wild pointer and refer the reviewers to the supplementary materials as it provides a better illustration.

---

### Official Review · Reviewer_Jac7 · 2022-07-27
**Well designed QA dataset**

**Rating:** 9
**Confidence:** 4
**Clarity:** This paper is well written and easy t…

**Strengths:**

- This paper introduces a new benchmark, EgoTaskQA that contains 40K balanced question-answer pairs. Those questions are targeted to understand the video from multiple perspectives to evaluate the agents’ intelligence.

- The questions are very broad, ranging from descriptive, predictive, counterfactual, and explanatory to evaluate the agent over the spatial, temporal, and causal understanding of the tasks.

- The generated questions in the proposed benchmark take care of the diversity and balance of each kind.

- The splits of normal and indirect of the dataset will help understand whether the model is using the correlation between the questions in the training and evaluation set without understanding the task, instead using language shortcuts in relations among questions.

- Evaluate model performance on question scopes, types, targeting semantics, and overall answer categories will show its overall capacity in understanding.

- The ablation of object information and language-only shows the objects are very important visual clues in QA tasks.


**Weaknesses:**

- The data scope is in-door goal-oriented tasks so this might cause limitations for a broader community.

**Additional Feedback:**

No.

**Correctness:**

The claims made are well supported by experiments in the paper. The evaluation methods and experiments design are appropriate and performed correctly.

**Documentation:**

This paper has sufficient details on QA data collection and organization, availability and maintenance, and ethical and responsible use. This paper includes documentation and intended uses in supplemental. It has a URL for reviewer access to the dataset and it has a hosting, licensing and maintenance plan. It has architecture and training details for reproducibility.

**Ethics:**

No.

**Relation To Prior Work:**

The difference from the previous benchmarks is shown in Tab.1.


**Summary And Contributions:**

This paper created the EgoTaskQA benchmark for a direct evaluation of question-answering on real-world egocentric videos. The designed questions are aiming for video understanding of the human tasks from different perspectives, action dependencies and effects, intents and goals, and agents’ beliefs about others.

This benchmark will help to evaluate agents capability in a comprehensive way with descriptive, predictive, explanatory ad counterfactual questions and thus help to develop more intelligent agents.

---

> ### Author Response · Authors · 2022-08-18
> **Response to the reviewer**
>
> We gratefully thank the reviewer for acknowledging the high future impact of this work and the clear summarization of our contributions. In response to the comments:
>
> ### 1. The data scope is in-door goal-oriented tasks so this might cause limitations for a broader community
> As discussed in the limitation section, we believe adding more diverse activities into EgoTaskQA is an important future step and will certainly increase its value as a general benchmark. With the development of large-scale benchmarks (e.g., there exists a point-of-no-return task in Ego4D [1]) and simulated environments (e.g., grounding instructions to states [2]), more and more world state information will be available including both indoor and outdoor scenes. We believe the quest for the world model just began and hope that our question generation pipeline will promote future research in this field. In the meantime, we will continuously refine our work to be a more general plug-and-play tool for world model studies.
>
> [1] Grauman, Kristen, et al. "Ego4d: Around the world in 3,000 hours of egocentric video." CVPR 2022.
>
> [2] Shridhar, Mohit, et al. "Alfred: A benchmark for interpreting grounded instructions for everyday tasks." CVPR 2020.

---

### Official Review · Reviewer_Peaw · 2022-07-28
**Overall great contribution to QA understanding in Ego Videos**

**Rating:** 7
**Confidence:** 3

**Strengths:**

Good contributions of augmenting datasets, novel Q/A for a video QA dataset (as described in table 1), good results and discussion, limitations described are relatively small, good figures in general (aside from grammatical issues described in weaknesses), normal/indirect dataset idea was a great idea and showed good results,

**Weaknesses:**

Poor grammar in benchmark vocabulary/spatial relations, certain figures' grammar is distracting from the paper (fig. 1 and 2), certain figures can be very difficult to understand due to complexity, doesn't explore model architecture/reasoning modules at all, lack of forward guidance for future papers/works, weak explanation for broader impact.

**Additional Feedback:**

The figures and figure captions need to be improved before publication for grammatical correctness as well as clarity. Also, the 3rd contribution mentioned of "devising challenging benchmarking splits over EgoTaskQA to provide a systematic evaluation" is unclear, it would be best to immediately mention the benchmark splits created (normal and indirect), as well as their purposes, rather than briefly mentioning them in the paragraph. Additionally, I was wondering if there was potential for mislabeled data as a cause (or one of many) for low benchmarking accuracy.

**Clarity:**

The text of the paper is pretty well written and clear. However, the figures have a lot of grammatical issues which makes apparent that the dataset likely also had substantial grammatical issues. These grammatical issues did at times reduce clarity.

**Correctness:**

The claims made in the paper are generally correct. There are concerns, however, on the correctness of relationships, grammar, and terms in the datasets augmented. As these datasets were one of this paper's main contributions, this is a relatively big issue. However, to my knowledge other claims made in the paper are correct and the benchmarking setup is correct.

**Documentation:**

Yes.

**Ethics:**

No.

**Relation To Prior Work:**

Yes, a table makes clear differences between this and other works as well as a comprehensive related works section.

**Summary And Contributions:**

Augmented two datasets based on Lemma dataset - a normal and indirect dataset. Both datasets had contributions of four question types (descriptive, predictive, explanatory, and counterfactual) across different question scopes (world, intents and goals, and multi-agent) from an egocentric view. The main difference between the indirect dataset from the normal dataset was the usage of words like "something" instead of object names to limit textual exploitation. They compared results between these two datasets and the indirect dataset results were worse than the normal dataset results, shedding light on the flaws of general SOTA video text alignment models as exploiting textual relationships rather than using sufficiently knowledgeable spatial temporal reasoning modules.

---

> ### Author Response · Authors · 2022-08-18
> **Response to the reviewer (2/2)**
>
> ### 3. Weak explanation for broader impact
> As we discussed in the related work, most existing intelligent robots depend on the understanding of world states to act and plan. With most existing works in video understanding focusing on the action, world states and their transition are less studied (though the community is showing emerging interests). We believe the state-transition annotation provided in the augmented LEMMA dataset can bridge between world-model learning in simulated environments and understanding real-world events with complex activities (e.g. multi-agent collaboration). It will also be a good testbed for learning tasks from real-world activity videos.
> For the proposed EgoTaskQA benchmark, we have also shown (as mentioned previously) that there exist challenges for existing models to conduct goal-oriented reasoning, and hope the provided questions can foster research in broader video understanding directions, as well as task planning and robotic applications.
>
> ### 4. The motivation and description for normal and indirect split
> We thank the reviewer for the suggestion. We will make the following changes in the revision to make the flow smooth: add a discussion on language information in goal-oriented question answering and motivate the design of direct/indirect split in L.51-52.
>
> ### 5. Potential mislabeled data as a cause for low benchmarking accuracy
> As mentioned in Sec.3.1, we used AMT for the annotation process and had verification rounds over the annotated state annotations. Therefore, we believe the mislabeled data, though exists, plays a minor role in the cause of low benchmarking accuracy. We will continue to address the minor issues of collected data in the future. Meanwhile, we notice that human performance is relatively low compared to other visual question-answering benchmarks (though significantly higher than current baselines). We argue that this indicates that identifying causal dependency between actions and doing multi-step reasoning for the rationales of actions (especially in a multi-agent partially-observable scenario) is also not a trivial task for humans, especially with indirect references to actions and objects. This further differentiates our work from existing video question answering benchmarks on detailed world state and task understanding, intent, as well as the multi-agent theory of mind.

---

> ### Author Response · Authors · 2022-08-18
> **Response to the reviewer (1/2)**
>
> We thank the reviewer for the insightful feedback and the positive comments on data curation, split design, and experimental results. The discussion of the raised questions is as follows:
>
> ### 1. Poor grammar in benchmark vocabulary/spatial relations
> We thank the reviewer for the suggestion on refining figures. As mentioned in the main text, our questions are programmatically generated from question templates. We further fill in different possible references for the missing blanks as illustrated in Sec.B.3 in the supplementary. During the preparation of the generation, we did our best to make the machine-generated question understandable to humans by designing proper templates. Though some of the generated questions are not strictly grammatically correct, we respectfully point out that the human evaluation of the questions does provide a reasonable estimate of whether these questions could be correctly understood by a human. The high human performance indicates that the potential grammatical flaws in questions are not the direct cause of the performance gap between humans and models. As we will continue refining our grammar templates, we hope that such problems could be relieved in future versions.
>
> ### 2. Exploration of model architecture/reasoning module, lack of forwarding guidance for future papers/works
> As discussed in the limitation section, the current experimental result is limited to the 6 state-of-the-art video-QA models, leaving the potential for better modules to be explored by future works. However, we do respectfully argue that the baselines provided in this paper are already sufficient as the initial evaluation. Specifically, the experiments demonstrate the usefulness of object information and language supervision in EgoTaskQA. With language pre-trained models taking a dominating role in current video-QA models, our results show the challenge of taming them for reasoning in a specific visual/language domain (especially from our indirect split experiments).
>
> More concretely, we plan to investigate the following two branches in the future, (i) explicit spatial-temporal grounding for modularized video-QA models, and (ii) prompting large-scale pre-trained models (both visual and language) for the task-specific video QA challenge. For (i), we believe our egocentric data, compared to third-person-view ones, will make the problem less ill-posed since it contains less occlusion on the interacting objects. We believe this feature consequently benefits the study of spatial-temporal localization and modularized model built upon grounded knowledge. As for (ii), we do recognize increasing efforts for adapting large-scale pre-trained models for reasoning [1] and video-QA [2], which are both insightful attempts for eliciting knowledge in pre-trained models. However, as our experiments suggest, adopting pre-trained models directly to a specific domain is non-trivial, especially with challenges in grounding (the significant drop of the ClipBERT model in indirect split). Therefore, how to let a large-scale pre-trained model reason in a specific domain will also be an interesting topic to study. With the additional available page, we will add a future work paragraph in Sec.5 to elaborate on these discussions.
>
> [1] Wei, Jason, et al. "Chain of thought prompting elicits reasoning in large language models." arXiv preprint arXiv:2201.11903 (2022).
>
> [2] Buch, Shyamal, et al. "Revisiting the" Video" in Video-Language Understanding." CVPR 2022.

---

### Official Review · Reviewer_qcQ6 · 2022-07-28
**Review of EgoTask QA**

**Rating:** 7
**Confidence:** 3

**Strengths:**

* Building upon an existing dataset considering what's missing in the existing works.
* The data annotation process is systematic and the authors provide the reasoning and procedures behind the annotation in the paper.
* Provides extensive evaluation of state-of-the-art models along with naive baseline and human performances.
* Provides detailed information including procedures of annotation, statistics, model setup, and datasheet in the supplementary materials.
* Has a website providing data exploration along with code and model checkpoints.

**Weaknesses:**

* The tables are hard to follow without bolding. For example, the highest numbers in Table 2 and the increase of performance in HCRN in Table 3 should be easier to read if bolded.
* Terms like "Most Likely" in Table 2 are vague and confusing. The authors should consider explaining more about the term to clarify.
* There are four columns but only two category columns in Table 3. The authors should consider adding an appropriate name for each column.

**Additional Feedback:**

* Can this annotation paradigm extend to non-ego-centric or goal-oriented video understanding tasks?
* What is the role of diagnostic datasets like the proposed work? It would be interesting to show the diagnosis of more models more extensively.

**Clarity:**

In general, the paper is well-written. However, the authors should consider clarifying terms and tables for the first-time readers.

**Correctness:**

The dataset construction process is systematic. The authors provide the procedure of the data annotation process. The evaluation methods are straightforward.

**Documentation:**

The documentation is fairly sufficient. Mentioned above.

**Ethics:**

Yes.

**Relation To Prior Work:**

Yes, prior work is well-cited with claims about the proposed dataset differing from prior works.

**Summary And Contributions:**

This paper introduces EgoTaskQA, a new benchmark with questions and answers for diagnostic analyses on spatial, temporal, and causal relationships between entities in goal-oriented task videos. The dataset is extended from the ego-centric multi-agent LEMMA dataset with four different types of QA annotations: descriptive, predictive, explanatory, and counterfactual. The paper also provides experiments with state-of-the-art models on the benchmark along with human performances for diagnoses on the reasoning tasks in goal-oriented task videos.

---

> ### Author Response · Authors · 2022-08-18
> **Response to the reviewer**
>
> We thank the reviewer for the insightful review. We appreciate the positive comments on the novelty of the benchmark, systematic annotation pipeline, extensive evaluation, and data visualization. We make the following clarifications to address the reviewer's concern:
>
> ### 1. Experimental result visualization (highlight numbers, make tables easier to read)
> We thank the reviewer for formatting suggestions. We will revise the paper and make the following changes: make the highest numbers bold in Tab.2, highlight the contrast in Tab.3 with different colors, and add highlighting and percentage drop in Tab.4 for better result visualization. We will also add names for each sub-column in Tab.3 (namely Acc. and Change) as requested.
>
> ### 2. Terms like "Most Likely" in Table 2 are vague and confusing.
> The "Most Likely" term refers to a simple heuristic-based question answering scheme where we collect all answers under each category in the training set and choose the most frequent one to answer questions for all test questions in the same category. This heuristic, compared to random predictions, can also show that our question-answer pairs are balanced, with no specific answer dominating the answer distribution. We will add the description to this simple baseline to Sec.4.1 L.242-243. Thanks for pointing this out.

---

### Review · Ethics_Reviewer_uxep · 2022-08-26

**Recommendation:** 1

**Ethics Documentation:**

It would be helpful and informative if the authors' would add a sentence or two in the paper explaining whether the subjects in the images consented to be in the images and whether there was any human subject oversight or supervision of the human subjects.
Because the authors of the instant paper are the authors of the paper related to the LEMMA dataset (2020) they appear to be in the best position to know whether consent was obtained from the participants in those images.

Assumptions of consent are not ethically sufficient, particularly as ethics concerns have grown over time. Consent was still required for human participation in human data models in 2019-2020.
If there are other reasons why the consent information is not available or cannot be obtained, an explanation in the paper or Appendix or Datasheet would be helpful.


**Ethics Review:**

The Ethics Review Guidelines used in this ethical review may be found at https://neurips.cc/public/EthicsGuidelines.

The LEMMA dataset used in the paper has been publicly available since at least July 2020. It is not readily available, but it appears that one may request access to it at https://sites.google.com/view/lemma-activity.
Many of the coauthors of the 2020 LEMMA dataset overlap with the instant authors. The LEMMA dataset was dataset was created to study goal-oriented task understanding in egocentric videos. The current work and dataset extended the LEMMA dataset with annotations consisting of object status, human-object and multi-agent relationships, and causal dependency structures between actions.
Actions with world-state transitions and their dependencies, agents’ intents and goals in task execution, and agents’ belief about others in collaboration were added to provide an in-depth evaluation metric for task understanding. These descriptive, predictive, explanatory, and counterfactual categories were used to systematically test models’ capabilities over spatial, temporal, and causal domains of goal-oriented task understanding.

Amazon's Mechanical Turk (AMT) workers were utilized as annotators. There does not appear to be any ethical issue raised by employing professional annotators paid to do the annotations. The use of AMT workers was disclosed in both the LEMMA and the instant papers.

The full image dataset does not appear to be available for review at this point. The authors note that it will be made available upon acceptance.
The dataset contains images of human subjects, but no information could be readily identified as to the informed consent of the subjects in the original LEMMA dataset or the present one. It would be helpful and informative if the authors' would add a sentence or two in the paper explaining whether the subjects in the images consented to be in the images and whether there was any human subject oversight or supervision of the human subjects. Because the authors of the instant paper are the authors of the paper related to the LEMMA dataset (2020) they appear to be in the best position to know whether consent was obtained from the participants in those images.

The dataset does not appear to be representative of the diversity of any given community. However, the actions of the individuals, rather than the individuals themselves appear to be the focus of the annotation and classification for intelligent agent situational understanding.
In the future, it may be helpful to include diversity in the datasets for actions that might be impacted by factors like age (young or old) or mobility.

The training of intelligent agents for situational understanding based on video or streaming poses ethical concerns, especially if used inside of a private residence. In this case, the 2020 LEMMA paper indicated the original dataset was curated inside AirBNBs.
The dataset could be used for intelligent agent surveillance and human rights concerns are raised when such intelligent agents could be used for monitoring vulnerable populations such as prisoners or in criminal profiling.
It would be helpful to others if the authors were to acknowledge their awareness of various use cases for their technology.

---

> ### Author Response · Authors · 2022-08-28
> **Response to the reviewer**
>
> We thank the reviewer for the comments and for thoroughly checking potential ethical issues of our paper. As requested, we added an additional paragraph in the supplementary Sec. E societal impact to discuss the agreement made by the participants of the video recordings.  We will update our revision on the supplementary.

---

### Author Response · Authors · 2022-08-29
**Paper revision submitted**

As the discussion phase is closing, we thank all reviewers for their efforts and constructive feedbacks. We have updated the draft and supplementary as promised (especially on figures, data curation/generation details, future work/forward guidance, and ethics clarifications) and welcome all feedbacks of the current vision. Feel free to leave a note :-)

---

### Meta-Review · Area_Chair_iaPx · 2022-09-02

**Recommendation:** Accept
**Confidence:** 5

**Metareview:**

The reviewers are positive regarding the high level of the contribution of the work for the NeurIPS 2022 Track Datasets and Benchmarks. The authors properly addressed all reviewers comments and concerns during the rebuttal period.

---

### Decision · Program_Chairs · 2022-09-16

Accept